# Field-programmable silicon temporal cloak

Feng Zhou[1], Siqi Yan[1], Hailong Zhou[1], Xu Wang[1], Huaqing Qiu[2], Jianji Dong [1], Linjie Zhou[3], Yunhong Ding [2], Cheng-Wei Qiu [4,5] & Xinliang Zhang[1]

Temporal cloaks have aroused tremendous research interest in both optical physics and optical communications, unfolding a distinct approach to conceal temporal events from an interrogating optical field. The state-of-the-art temporal cloaks exhibit picosecond-scale and static cloaking window, owing to significantly limited periodicity and aperture of time lens. Here we demonstrate a field-programmable silicon temporal cloak for hiding nanosecond-level events, enabled by an integrated silicon microring and a broadband optical frequency comb. With dynamic control of the driving electrical signals on the microring, our cloaking windows could be stretched and switched in real time from 0.449 ns to 3.365 ns. Such a field-programmable temporal cloak may exhibit practically meaningful potentials in secure communication, data compression, and information protection in dynamically varying events.

---

[1] Wuhan National Laboratory for Optoelectronics, School of Optical and Electronic Information, Huazhong University of Science and Technology, 430074 Wuhan, China. [2] Department of Photonics Engineering, Technical University of Denmark, DK-2800 Kongens, Lyngby, Denmark. [3] State Key Laboratory of Advanced Optical Communication Systems and Networks, Depart of Electronic Engineering, Shanghai Jiao Tong University, Shanghai 200240, China. [4] Department of Electrical and Computer Engineering, National University of Singapore, Engineering Drive 3, Singapore 117583, Singapore. [5] NUS Suzhou Research Institute (NUSRI), Suzhou Industrial Park, Suzhou 215123, China. Correspondence and requests for materials should be addressed to J.D. (email: jjdong@hust.edu.cn) or to C.-W.Q. (email: chengwei.qiu@nus.edu.sg) or to X.Z. (email: xlzhang@hust.edu.cn)

Since the breakthrough concept of temporal cloak was proposed by McCall et al. at the first time[1], a great deal of attention was paid to new methodology and experimental verification of temporal cloaks[2–13]. Particularly, the concept of temporal cloak first came true with an experiment by Gaeta et al.[14], and later Weiner et al. facilitated temporal cloak into more practical applications at telecommunication data rate with superior cloak efficiency[15]. Temporal cloak, derived from spatial cloaking due to the space-time duality[16,17], does not create a void to conceal spatial objects[18–20], but open and suture a time gap to hide events in time-domain. To create a time gap as the cloaking window, a time lens with quadratic phase profile is required to bend the light along the time and a dispersive medium is used to converge the light beam to a time spot[14,15,21]. The suture of the time gap is implemented by an opposite dispersive medium. A lot of temporal cloak approaches were proposed theoretically and confirmed to be feasible in experiments. However, the existing cloaking experiments exhibited only a fixed and small cloaking window with picosecond-level due to the periodicity and aperture limit of time lens. In early years, the time lens was created by a pair of split time lens employing four-wave mixing in a highly-nonlinear fiber (HNLF)[14], suffering from strong pump consumption and low energy efficiency. Afterwards, the time lens was created by temporal Talbot effect employing phase modulator (PM) and chirped fiber grating[15,21]. Unfortunately, a PM could only provide very limited aperture of time lens when driven by a sinusoidal voltage[22]. Even the PM was optimized by chirped component and cascaded structure, the maximum continuously cloaking windows was 196 ps, far from nanosecond-level. A large cloaking window has been pursued all along since it represents the hidden capacity available in a time slot for secure communications. For some specific scenario of secure communication, data protection is an important secure device, allowing for sharing some public data to the user but concealing other private data in real time. Thus, it is very important to make cloaking window field-programmable (i.e., the cloaking window can be switched off, switched on, or stretched freely, similar to the concept of field programmable gate arrays in digital circuits) since different types of optical packets can be hidden at any time slots with the cloaking system. However, all state-of-the-art cloaking systems so far featured periodical cloaking windows without field-programmability.

In this paper, we demonstrate a field-programmable silicon temporal cloak with a record cloaking window at the nanosecond-level, benefiting from a unique electrically controllable silicon-based time lens. The superior time lens consists of an optical frequency comb and an electrically tuned microring resonator (ET-MRR) acting as a scanning filter, whose output wavelength is proportional to the applied voltage. The electrically controllable time lens is enabled by applying an electrical split sawtooth signal on the ET-MRR and disabled by applying a direct current (DC) electrical signal. This electrically controllable silicon-based time lens has distinct advantages of field-programmable cloaking window, moderate power consumption and compact photonic integration. To break the periodicity of the cloaking window, we demonstrate, for the first time, a field-programmable silicon temporal cloak with potential applications in data protection, enabling to share some public data to the user but conceal other private data in real time. In addition, we obtain a record cloaking window of up to 3.365 ns, which is 17 times larger than the longest time window reported so far[21]. Furthermore, we succeed in concealing pseudorandom dark return-to-zero (RZ) signal at a rate of 200 Mbit/s and verify the stretchability of cloaking window from 3.365 ns to 0.449 ns by changing the repetition rate and peak voltage of driving electrical signals. We suggest that the real-time programmability of temporal cloak may make its applications, such as secure communication and data compression, more practical and closer to our daily life.

## Results

**Operation principle**. Time-lens is a core of temporal cloak, which determines the capacity of cloaking window at the event plane. To intuitively understand the cloaking window, the probe light waveform at the event plane is illustrated in Fig. 1a. The zero-intensity regions represent the time gaps as the cloaking windows[23]. Within these time slots, any temporal events will be concealed and not appear to the observer. In the traditional temporal cloaking systems shown in Fig. 1a, the open windows can only last for <200 ps[14,15,21,24] because of the limited aperture of time-lens. Besides, the open windows are periodic, which means all events in the cloaking windows are concealed periodically without flexibility.

Figure 1b shows the probe light waveform at the event plane in our cloaking scheme, featuring a large cloaking window (~3 ns) and the cloaking window can be field-programmable. The large cloaking window makes it possible to conceal optical packets with nanosecond-level events, and the controllable cloak windows allow for real-time switching off, switching on, and stretching the cloaking window freely. The field-programmable temporal cloak enables to share some public data to the user (at the state of *cloak off*) but conceal other private data (at the state of *cloak on*, gray labels) in real time, similar to a hardware of data protection. However, the traditional temporal cloaking systems were unable to do so.

Figure 1c describes the schematic of the proposed field-programmable silicon temporal cloak, and the bottom insets show the waveforms and spectra of the light transmission, electrical control signal and event at different locations in the cloaking system. A continuous wave (CW) light (Fig. 1d) is firstly converted into a broadband flat optical frequency comb source via an optical frequency comb generator (Fig. 1e, green), then launched into the ET-MRR. The ET-MRR acts as a swept-frequency filter when driven by an electrical split sawtooth waveform (Fig. 1f, green curve within gray label). The ET-MRR is fabricated on a commercial silicon-on-insulator (SOI) wafer consisting of a ring waveguide and two straight waveguides. The microscope images of the fabricated ET-MRR and the zoom-in ring region is shown in the top inset. The details of ET-MRR fabrication can be found in Methods section for device fabrication. The detailed scheme and the whole micrograph of the ET-MRR can be found in Supplementary Note 1. The output wavelength dependent on time is a split sawtooth function (Fig. 1g, pink) curve within gray label since the output wavelength of optical frequency comb source is continuously scanned by the swept-frequency filter. Then, the electrically controllable time lens is created. The detailed principle of the electrically controllable time lens can be found in Supplementary Note 2.

While the cloak is set to *cloak on* (i.e., the ET-MRR is driven by a split sawtooth waveform, gray labels), the light then passes through a dispersion element, such as a spool of single mode fiber (SMF), where the shorter wavelength light propagates faster than the longer wavelength light. Thus, the dispersion effect makes energy of longer wavelengths and shorter wavelengths converge into temporal pulses and leave a time gap, which is shown in Fig. 1h (blue curve within gray label), and the cloaking window (or time gap) is opened. Any event during this time gap will not be perceived by the probe. When the probe light experiences an opposite dispersion amount, such as a spool of dispersion compensating fiber (DCF), the cloaking window will be sutured. Then the probe light is detected by a photodetector (PD) and the corresponding output waveform is observed by an oscilloscope, as

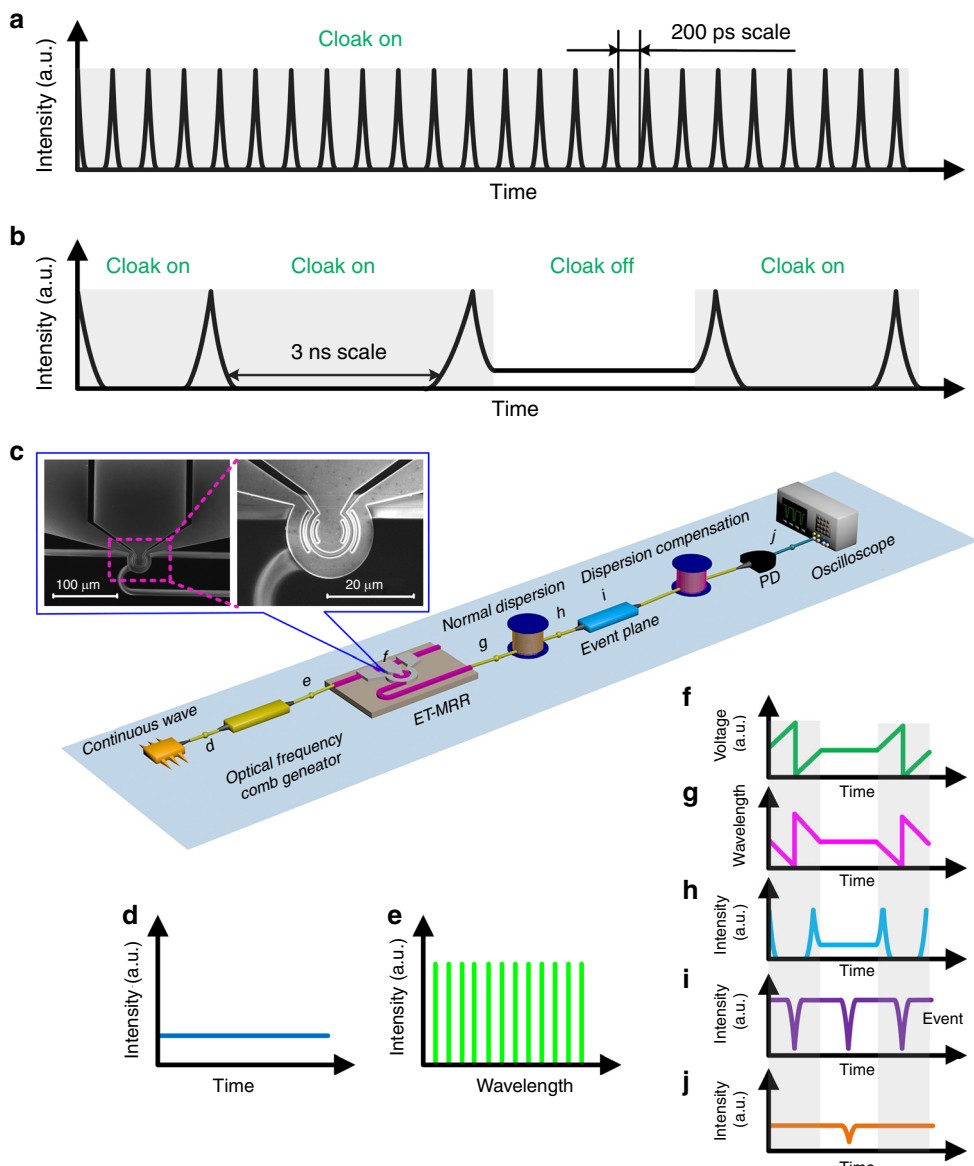

**Fig. 1** Schematic and representative of field-programmable silicon temporal cloak. **a** The probe light waveform at the event plane of the traditional temporal cloak. **b** The probe light waveform at the event plane in our cloaking scheme, featuring a large cloaking window (~3 ns) and the cloaking window can be field-programming freely. **c** Schematic of field-programmable silicon temporal cloak. d-j represent various points in the circuit and the characteristic curves corresponding to the probe light, electrical control signal and event are as follows: (**d**) the probe light is continuous wave; (**e**) optical frequency comb generator creates a broadband flat optical frequency comb; (**f**) an ET-MRR driven by an electrical split sawtooth signal and DC signal; (**g**) the output of drop-port of ET-MRR is split linearly chirped optical signal or monochromatic light, which is required for field-programmable temporal cloak; (**h**) opening gap is achieved while the ET-MRR is driven by a split sawtooth waveform (*cloak on*), and temporal gap does not open while the ET-MRR is driven by a DC signal (*cloak off*); (**i**) event (dark RZ); (**j**) the output waveform is CW (*cloak on*) or inverted pulse (*cloak off*). Inset: the microscope images of the fabricated ET-MRR and the zoom-in ring region. The gray labels represent the cloaking bits

shown in Fig. 1j (orange curve within gray label). For the observer, the signal waveforms at input and output planes are the same, and the events could not be perceived. While the cloak is set to *cloak off* (i.e., the ET-MRR is driven by a DC voltage), the output optical field of the electrically controllable time lens is monochromatic light. Whether the probe light passes through a normal dispersion element or an opposite dispersion amount, the light waveform remains essentially unchanged and time gap disappears. In this case, the events can be captured by the probe light and the corresponding output waveform is shown in Fig. 1j (orange curve without gray label). Therefore, this superior time lens can alter the cloaking window in real time and produce the

field-programmable silicon temporal cloak. In order to describe the cloaking system in details, we also simulate all the cloaking states of the cloaking system (see Supplementary Note 3). Note that the wavelength has changed when the probe light passed through the cloaking system. Thus, this hiding event may be still scouted from the spectral evolution. To solve this issue, a wavelength converter can be inserted after the dispersion compensation to convert multi-wavelength signal to a single wavelength as same as the input light[25,26]. In this way, the output signal of our system will turn to be a continuous wave light and the cloak could function not only in the time domain but also in the frequency domain, as a full temporal cloak.

**Electrically controllable silicon-based time lens**. To demonstrate the proposed cloaking system, the crucial element is the implementation of the electrically controllable silicon-based time lens, which consists of an optical frequency comb generator and an ET-MRR. The tuning mechanism of ET-MRR is based on the plasma dispersion effect. The measured transmission spectra at the drop-port of the ET-MRR are illustrated in Fig. 2a, when different DC voltages are applied on it. One can see that as the applied voltage increases, the Q factor decreases, and the resonance wavelength experiences blue shift due to carrier dispersion effects. We also notice that when the driving voltage increases, the ET-MRR transmission at the peak would decrease (see Supplementary Note 4). The difference of the transmission peak is up to 10 dB, when the driving voltage varies from 0.8 to 1.3 V. This difference may deteriorate the cloaking performance without spectral compensation. To compensate the uneven transmission of ET-MRR when blue shifted, a clival optical frequency comb generator is employed. Here we use a PM together with a wave shaper to generate a clival optical frequency comb. A low-$V_\pi$ PM is driven by a sinusoidal radio frequency (RF) signal with a repetition rate of 10 GHz. Moreover, this sinusoidal RF signal is strongly amplified to about 30 dBm. A broadband flat optical frequency comb (green) is generated as shown in the inset of Fig. 2b. We further use a wave shaper (see the inset of Fig. 2b, purple) to tailor the flat optical frequency comb to a clival optical frequency comb with 10 dB difference of the peak power, as depicted in Fig. 2b, which exactly covers the sweeping spectral range of the ET-MRR. Finally, an electrical signal (mixed split sawtooth signal and DC signal) with repetition rate of 200 MHz (Fig. 2c) is applied to the ET-MRR. The electrically controllable silicon-based time lens is constructed while the ET-MRR is driven by a split sawtooth waveform, and time lens does not work while the ET-MRR is driven by a DC signal. The instantaneous frequency of the output signal of the ET-MRR (Fig. 2d) is calculated according to the characteristic curve of wavelength shift (Supplementary Fig. 4a) and the electrical signal (Fig. 2c). Note that

the finite width of the curve in the y-direction is attributed to the bandwidth of the ET-MRR.

**Experimental demonstration**. According to Fig. 1c, we verify the experiment of a field-programmable temporal cloak. The detailed experimental setup can be found in Supplementary Note 5. In our experiment, the cloaking window can be easily opened and closed by setting the split sawtooth and DC signal applied on the ET-MRR, respectively. Without loss of generality, the split sawtooth signal is recorded as cloaking bit 1, representing the state of *cloak on*, and the DC signal is recorded as cloaking bit 0, representing the state of *cloak off*. Figure 3 shows some typical experiment results of field-programmable temporal cloak by varying the cloaking bits on the ET-MRR. The first row shows the sequence of cloaking bits at the bit rate of 200 Mbit/s. The second row shows a user-defined sequence event (blue) with dark RZ signals. The third row shows the output waveforms controlled by the cloaking bits. To characterize the performance of the temporal cloak, we define the ripple factor as the ripple voltage over the average voltage of output signals in the state of *cloak on*. Ideally, in a perfect temporal cloak, the ripple factor should be as small as possible. In Fig. 3a, the output waveform is a constant with a low ripple factor (~30%), which means all event bits have been hidden successfully since all the cloaking bits (cyan) on the ET-MRR are set to 1111111111. From Fig. 3b, c, the switching bits (cyan) is set as random sequences, such as 0101100111, and 1100111000, respectively. Then we can observe the event waveforms at cloaking bit of 0 (bright region), but cannot observe the event waveforms at cloaking bit of 1 (gray region). The ripple factor is about 25% in the gray region, representing a good cloak performance. Figure 3d shows that the output waveform is the same as the event waveform when the cloaking bits are all set at 0s, representing the state of *cloak off*. The output signal is broadened after the DCF and it can be compensated by introducing a spool of SMF to compensate the dispersion of the DCF. Further, we set

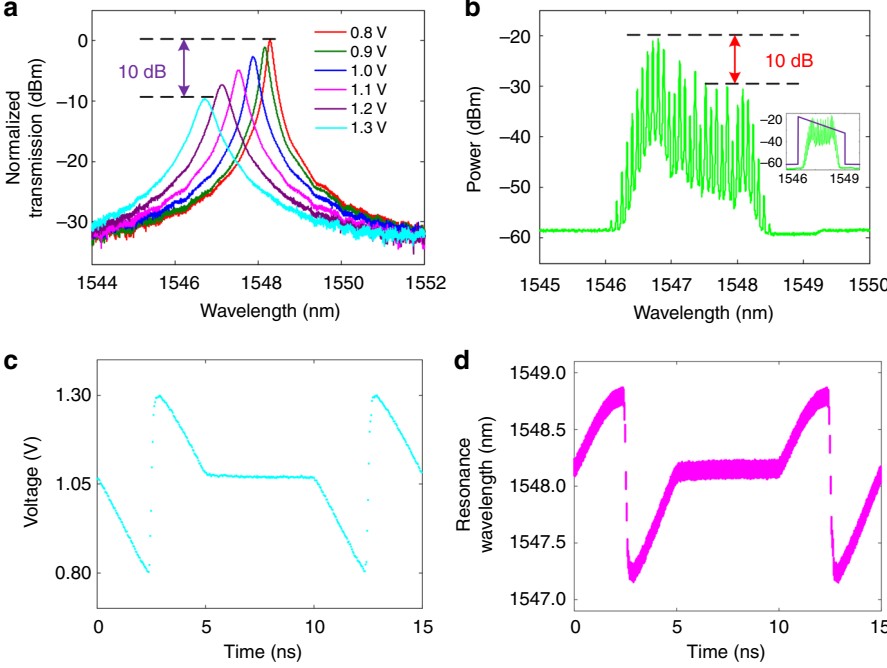

**Fig. 2** Electrically controllable silicon-based photonic time lens. **a** Measured transmission spectra at the drop-port of the fabricated ET-MRR with different DC voltages applied on the microelectrode. **b** The output spectrum of the optical frequency comb generator. Inset: original flat optical frequency comb (green) and optical filter of wave shaper (purple). **c** Driving electrical signal (mixed split sawtooth signal and DC signal) of the ET-MRR. **d** The time-resonance wavelength curve of the time lens is calculated

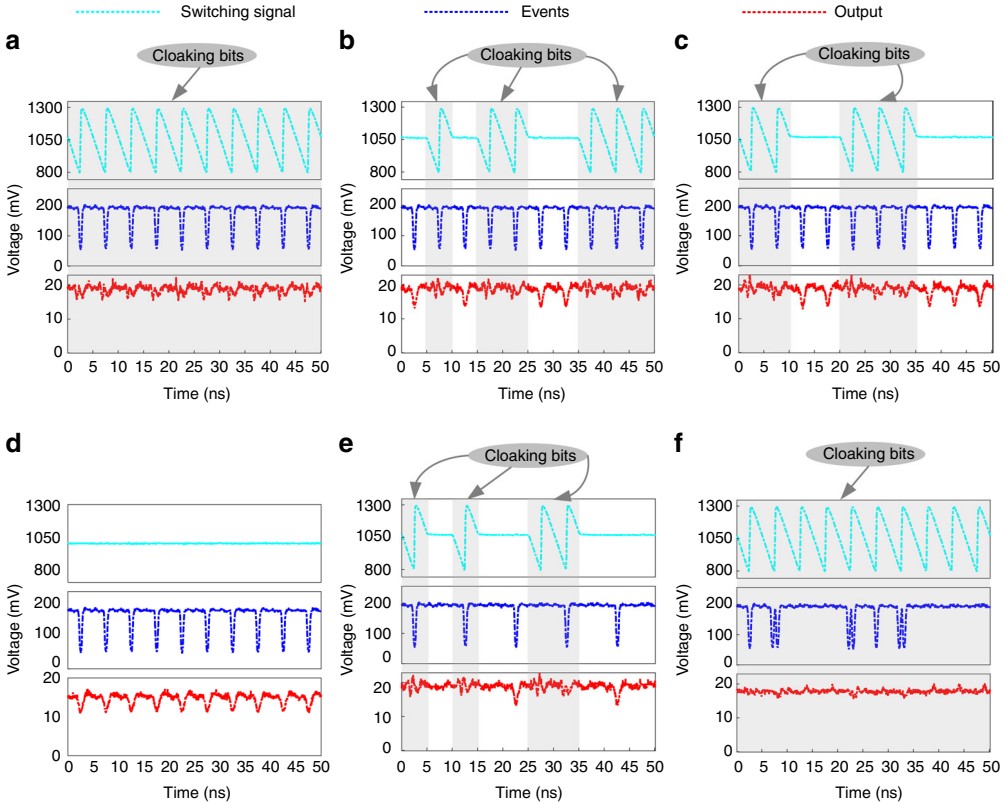

**Fig. 3** Field-programmable temporal cloak. **a** The DC waveform (red) with a low ripple factor is received when the cloak switch (cyan) and events (blue) are turned on all the time. **b**, **c** The received output waveforms (red) with high-level signal at the cloaking bits when the events (blue) are all turned on and the cloak switch (cyan) is selectively turned on. **d** The high-contrast voltage (red) output waveform when the cloak switch (cyan) is turned off and the events (blue) are turned on all the time. **e** Arbitrary sequence temporal cloak for hiding periodic sequence dark events with half cloaking repetition rate. **f** Periodic temporal cloak for hiding arbitrary sequence dark events. The gray labels represent the cloaking bits

the switching bits to random sequences such as 1010011000, and select 100 MHz dark RZ signal as the event. As shown in Fig. 3e, the periodic sequence dark events with half cloaking repetition rate is selectively cloaked by the arbitrary sequence temporal cloak. In addition, an arbitrary sequence dark events with both two-pulse event and one-pulse event are erased by the periodic temporal cloak, as shown in Fig. 3f. It shows that the long cloaking window could hide more pulses, not only a single pulse. Therefore, the field-programmable cloak has been successfully demonstrated. We may notice that the waveform contrast between *cloak on* and *cloak off* is not very high. In our scheme, the electrically controllable silicon-based time lens with a swept-frequency filter introduces a high loss for the output light signal. When the broadband optical frequency comb propagates through the swept-frequency filter and the selected light signal with very low power is amplified by the erbium doped fiber amplifier (EDFA), some noise is introduced. Thus, the signal-to-noise ratio is not high. The waveform contrast can be improved further by optimizing the DCF length, link loss, and the electrical signal on ET-MRR.

In order to more clearly characterize the cloaking window of our temporal cloak, we create periodical temporal cloak by setting the sawtooth voltage with repetition rate of 200 MHz applied on the ET-MRR. Figure 4a shows the opened time gap waveform with a repetition rate of 200 MHz that is measured at the event plane in our experiment system. We define the continuously cloaking window as the duration where the intensity of the probe drops below the 10% of the peak voltage, which is measured as 3.365 ns, 17 times larger than the previous experimental record (196 ps)[21]. Then the temporal event is emulated with a dark RZ

pseudo-random binary sequence (PRBS) at a bit-rate of 200 Mbit/s and its temporal waveform is detected by a PD with 5 GHz bandwidth, as shown in Fig. 4b. There are 2 levels in the eye diagram, caused by the electrical arbitrary waveform generator (Keysight M8195A) without calibration. The 5 GHz-bandwidth PD aims to suppress the beating signals from the optical frequency comb source. More details can be referred in the Supplementary Note 5. Subsequently, we measure the temporal waveforms at four different states of the temporal cloak, as shown in Fig. 4c, d. When the event is turned on and the cloak is turned off, we can capture the event signals with high-contrast voltage (Fig. 4c, blue). The measured eye diagram shows a thick eyelid which is caused by the high loss of the fiber link and photonic chip, as well as the fiber amplifier noise. And once the cloak is turned on (the event is kept on), the event signals are hidden and one can only observe a nearly continuous waveform (Fig. 4c, red). The ripple factor is ~13.33%. Note that the level is also different between *cloak on* and *off* because of amplifier saturation of the EDFA, which keeps the average output power constant. Thus, the CW waveform drops slightly compared to the modulated case. At the same time, the non constant higher level and the ripple is caused by the mismatch between the chirp of the time lens and the dispersion devices. If the event is turned off, whether cloak is turned on (Fig. 4d, black) or off (Fig. 4d, pink), the temporal output is the same continuous waveform, and the ripple factors is ~12.33 and ~6.67%, respectively. It indicates that a high quality of temporal cloak is successfully implemented. In addition, we also investigate the pre-compensation of wave shaper, which follows the optical frequency comb generator. If the wave shaper is removed, for the state of *Event off and Cloak on*,

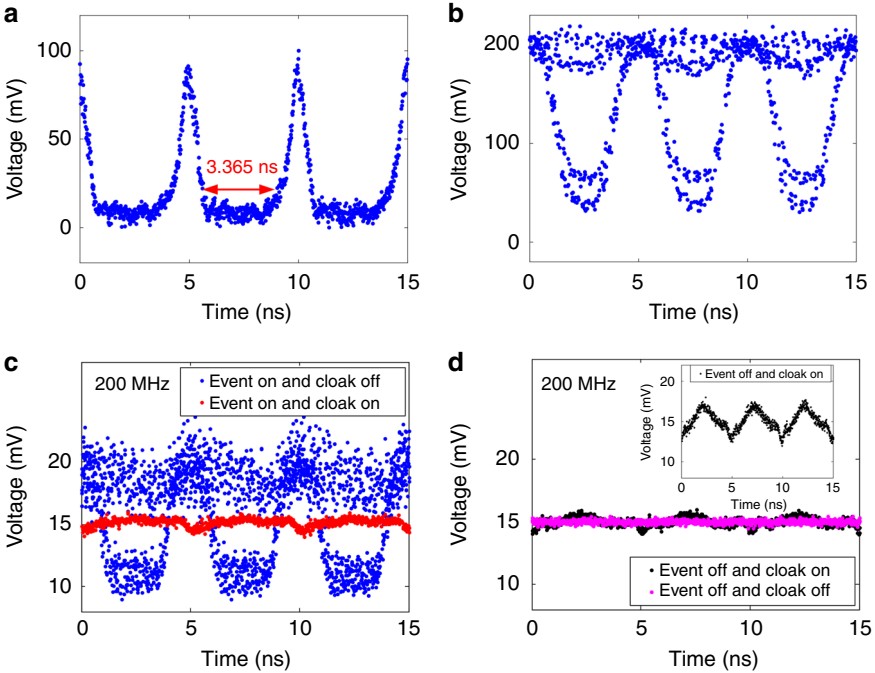

**Fig. 4** Experimental results of a nanosecond-event temporal cloak. **a** Measured waveform at the event plane. The continuously cloaking window is up to 3.365 ns. **b** Original event, a dark RZ at a bit-rate of 200 Mbit/s. **c** When the event is turned on and the cloak is turned off, the high-contrast voltage (blue) is received at the end of the circuit. And once the cloak is turned on (the event is kept on), the event signal is reduced to a continuous waveform (red). **d** If the event is turned off, whether the cloak is turned on (black) or off (pink), the temporal output is the same continuous waveform. Inset, the temporal waveform is measured at the end of the circuit, while the cloak is turned on, the event is turned off and the wave shaper is removed

the captured waveform has a large ripple (see the inset of Fig. 4d). The ripple factor is as high as 40%, indicating a poor performance for the cloaking system. Thus, the wave shaper can greatly reduce the ripple factor and improve the cloaking performance.

A stretchable cloaking window is of importance for a temporal cloaking system since different duration of optical packets may be hidden in the cloaking system. In our experiment, the cloaking window can be easily altered by changing the repetition rate and peak voltage of the ET-MRR drive signal. The reason lies in that the width of the cloaking window is determined by the amount of the chirp of the time lens and the dispersion in our cloaking system. The amount of the chirp of the time lens depends on the driving voltage of the ET-MRR. Thus, we can change the width of the cloaking window by changing the driving voltage of the ET-MRR while keeping the fiber at the same length. Figure 5a, b show the measured cloaking window waveforms at the event plane when the repetition rate of sawtooth signal is 400 and 800 MHz, respectively. The continuously cloaking window (time gap) is measured to be 0.979 and 0.449 ns, respectively. When we turn on both the states of cloaking and event, the measured output waveforms are shown in Fig. 5c (red, 400 MHz) and 5d (red, 800 MHz). The ripple factor is about 20 and 27%, respectively. These nearly constant intensity waveforms mean that the events are successfully hidden. When the cloak is turned off, the clear eye diagrams are shown in Fig. 5c (blue, 400 MHz) and 5d (blue, 800 MHz). It means the hidden events appear again after the cloaking is turned off. Therefore, our cloaking system can stretch its cloaking window easily. Note that the eye patterns for bit rate of 400 and 800 MHz are worse than that of 200 MHz, because the aberration of time lens is more serious at higher operation speed, limited by the time response of the ET-MRR.

## Discussion

Our field-programmable silicon temporal cloak is based on an electrically controllable time lens, which is compared to the state-of-the-art temporal cloak technologies with experimental verifications. The performance comparison is summarized in Table 1. The split time lens-based temporal cloak with HNLF was limited by the low total cloaking ratio and low bit rate[17]. Afterwards, the performances of the temporal cloak were greatly improved by using temporal Talbot effect[6,15], inverse temporal Talbot effect[21] and ultrashort pulse compression with PM[24], but these cloaking systems featured periodical and small cloaking windows without flexibility. In contrast, in our scheme, the cloaking window can be switched off, switched on, and stretched freely by setting the corresponding control signal of the swept-frequency time lens. Thus, our temporal cloak offers a field-programmable cloaking window, which is able to share some public data to the user but conceal other private data in real time at telecommunication data rate. Especially, the cloaking window is greatly extended to nanosecond-level.

Meanwhile, photonic integration of the cloaking system is also desirable with greatly reduced size and weight compared to the conventional discrete component approaches. This work represents a key step forward since the electrically controllable swept-frequency time lens has great potentials to be fully chip-integrated. For example, in terms of higher density chip-integration, the swept-frequency time lens can be potentially implemented by incorporating a Kerr comb (i.e., nonlinear MRR)[27–35] with swept-frequency filter (i.e., ET-MRR) on a common silicon nitride platform. Second, the cloaking window for cloaking system is mainly determined by the modulated bandwidth of ET-MRR (typical less than 1 GHz) and dispersion amount of integrated dispersion device. Third, an additional 5 GHz-bandwidth PD is used to suppress the beating signal but restrains the modulated rate of event. If a high Q factor ($>10^5$) MRR is used with very narrow bandwidth in the future[36–40], then the PD bandwidth has no limitation. Fourth, a wave shaper is used to tailor the spectral shaper of optical frequency comb source, at the

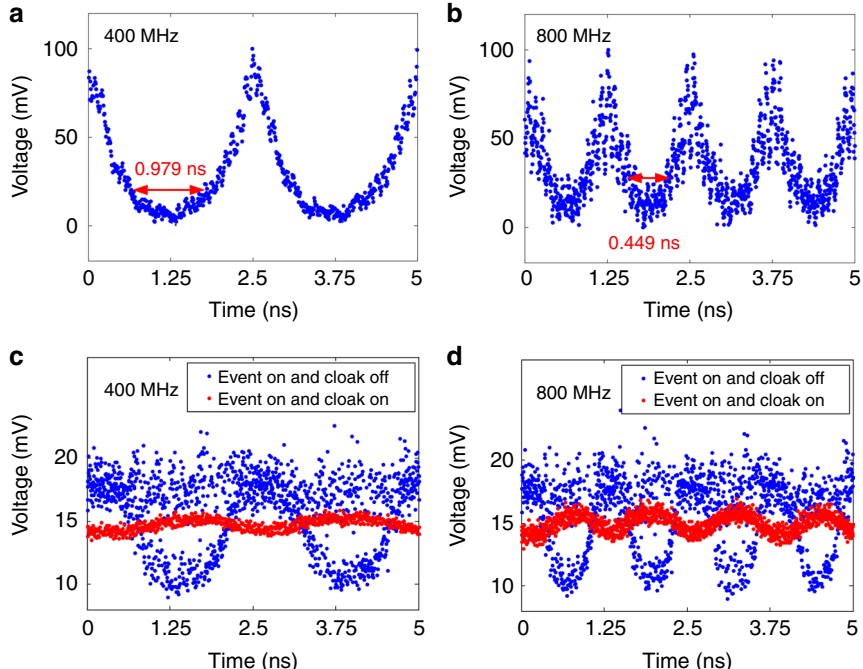

**Fig. 5** Stretchable cloaking window. **a**, **b** The received cloaking window waveform with the time gap of 0.979 and 0.449 ns at the repetition rate of 400 and 800 MHz, respectively. **c**, **d** Received signals when the event is turned on and the cloak is turned off (blue) or on (red) at the cloaking repetition rate of 400 and 800 MHz, respectively

**Table 1 Comparison of temporal cloak experiments**

| Ref | Mechanism | Key element | FP | CCW (ps) | Photonic integration |
|-----|-----------|-------------|----|----|---------------------|
| 14 | Split time lens | HNLF | No | 50 | No |
| 15 | Temporal Talbot effect | PM | No | 36 | No |
| 6 | Temporal Talbot effect | PM | No | — | No |
| 24 | Pulse compression | PM | No | 88 | No |
| 21 | Inverse temporal Talbot effect | PM | No | 196 | No |
| This work | Swept-frequency time lens | ET-MRR | Yes | 3365 | Yes |

*FP* field-programmable, *CCW* continuously cloaking window

cost of high loss of optical link. A possible solution is to adopt a thermally tuned microring resonator (TT-MRR) with a moderate modulated rate[41–43]. TT-MRR can offer a constant transmission as a swept-frequency filter, and a more linear frequency chirp for the time lens, compared to ET-MRR. Finally, to fabricate the entire cloaking system on a silicon chip, an on-chip high-speed intensity modulator is required, which acts as the event emulator. Currently, the integrated high-speed intensity modulator is mature in the SOI fabrication foundry[44–46], which motivates a promising future for the full integration.

In conclusion, we demonstrate the field-programmable silicon temporal cloak, benefiting from a unique electrically controllable silicon-based time lens. The superior time lens is enabled by an electrical split sawtooth signal applied on the ET-MRR and is easily disabled by a DC signal. We demonstrate a field-programmable silicon temporal cloaking system that break the periodicity and can be used for data protection. Furthermore, the recorded cloaking window up to 3.365 ns is obtained and the stretchability of cloaking window from 3.365 to 0.449 ns is verified by changing the repetition rate and peak voltage of driving electrical signals. The programmability of temporal cloak may make its applications, such as secure communication and data compression, more practical and closer to our daily life.

## Methods

**Devices fabrication**. We employ an add-drop ET-MRR to implement the field-programmable temporal cloak. Firstly, we design and fabricate the MRR on a commercial SOI wafer with 220-nm-thick silicon device layer on 2-μm-thick buried oxide layer. Secondly, we employ deep ultra-violet photolithography using a 248-nm stepper to define the waveguide patterns, followed by anisotropic dry etching of silicon. Thirdly, we implant boron and phosphorus ion implantations to form the highly *p*-type and *n*-type doped regions. In addition, the slab layer is etched outside the *p-i-n* junctions to confine the current flow around the ring waveguide. Finally, contact holes are etched and aluminum is deposited to form the metal connection. We use vertical grating coupling method to couple the light to the silicon MRR. The whole fabrication process is completed using complementary metal-oxide semiconductor compatible processes.

## Data Availability

The data that support the findings of this study are available from the corresponding author upon reasonable request.

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

## Acknowledgements

The work is supported by the National Natural Science Foundation of China (No. 61475052, 61622502, 61571186) and the Shanghai Municipal Science and Technology Major Project (2017SHZDZX03).

## Author Contributions

J.D., F.Z. and C.Q. conceived the idea. L.Z. designed and fabricated the electrically tuned microring resonator. J.D., F.Z. and C.Q. designed the experiment. F.Z. performed the measurements. J.D., F.Z., C.Q., S.Y., H.Z., X.W. and H.Q. discussed and analyzed the data. X.Z. and J.D. supervised the project. F. Z. wrote the first draft of the manuscript. All authors contributed to the discussion of the results and the writing of the manuscript.
