## [Peer Review File · Nature Communications]

Reviewers' comments:

Reviewer #1 (Remarks to the Author):

The author shows a temporal cloaking scheme similar to the split time-lens method with few differences and the main one is that the chirping is done by a tunable micro-resonator fabricated on a silicon wafer. This allows controlling on the opening and the closing of the cloaking in real-time.

The progress of temporal cloaking is fast, and indeed this idea is important and can lead to the implementation of temporal cloaking in telecommunication systems. However, several points need to be addressed for this paper to be ready for publication.

1. There are several problems in the cloaking concept presented here that must be addressed. First, in a temporal cloaking, like the one presented in ref. 14, the system starts with a specific CW wavelength and after the gap opened and closed, it returns to the same wavelength, state of polarization, and mode. Thus, a detector at the end cannot tell that the event happened nor he can tell that there was a cloaking device. Here, the wavelengths do not go back to the original beam. The generated frequency comb is not balanced by an opposite device which changes them beam back to the original wavelength. This is an important point which limits the proposed idea from being implemented in some systems and it must be addressed.
2. The author must emphasize that when changing the width of the cloaking window, the lengths of the dispersion fibers must be changed as well.
3. I had a hard time understanding the concept. The author does not explain well the working principle of the system. Even after reading the supplementary materials, it took me great effort to fully get it.
 - a. The author should spend more time explaining the principles of the ET-MRR. In line 59 for example, when the author state "The electrically controllable time lens is enabled by applying an electrical split saw-tooth signal on the ET-MRR and disabled by applying a direct current electrical signal." It should continue with explaining what is the influence of the electric field on the light. Specifically, I think that it should be: "which acts as a tunable filter were its output wavelength is proportional to the applied voltage"
 - b. A scheme of the ET-MRR is missing with an explanation of what are all the waveguides are for and which waveguide is coupled to which and where. The microscope picture of the device is far from being enough.
 - c. Fig 1(a) and (b) are nice and nowadays it is very common to have a 3d system drawing as is shown in (c) although there is no benefit in such a drawing over a simpler 2d drawing. But the problem is in (d). Since the system is drawn in 3d, all the graphs are tilted which makes them hard to compare and understand. I believe that these graphs should be oriented flat on the page. And maybe even combine several curves in a single graph. For example, curves IV, V, and VI should be on a single graph. While the rest can have a different graph for each curve.
4. Fig. 2(d) should not be drawn with a thin curve since the ET-MRR has a 0.4 nm width, this should also be the width of the curve in the y-direction. This will help in explaining the limitations of the system.
5. Can the author suggest a way to fabricate the entire cloaking system on a silicon chip?

Once these comments are addressed, I believe that this manuscript can be accepted for publication.

Reviewer #2 (Remarks to the Author):

The paper from Zhou and coworkers deals with the experimental demonstration of temporal cloaking based on the technique of time lens. In their work, they used a novel device made of a programmable electrically tuned micro-ring resonator to create the chirp profile involved in the time lens operation. The paper is well written and provide a pedagogical approach. However, to my opinion, the novelty and quality of the results are not good enough to deserve publication in a

high impact journal, in particular Nature Communications. Consequently, I reject this paper and encourage the authors to submit their work elsewhere in a less impact journal after the following revisions.

1. The concept and physical effects are not new and was already demonstrated in various publications, in particular in [14,15,21] but not only.
2. The only novelty is the implementation of a programmable silicon micro-ring. However, the time response appears very low compare to phase modulator. It is probably the reason why the authors focus the paper on the record temporal gap they succeed to open. The advantage of their device is the programmable nature but it is not very clearly demonstrated to my opinion. To really support their claims, the authors should have demonstrate some cloacking with arbitrary sequences and duration of the cloacking window, not only some periodic train as provided by figs. 4 and 5. Indeed, the demonstration and advantage of this programmable device is almost absent of this paper, except in figs. 3c-e but for periodic dark event. Moreover, the authors should have demonstrated the cloacking of data packets at high repetition rate rather than very low repetition rate data, 200Mbit/s is very week. Indeed, the advantage of generating long temporal gap could be to hide long data packet, not only a ns pulse.
3. The way the authors qualified the cloacking efficiency is quite questionable. An extinction ratio between ON and OFF should be better. 2 digits is also unnecessary. Moreover, the authors used a very large scale to present their results of cloacking. In fig. 3 for instance, it is difficult to see and fairly judge the efficiency of the process. It seems to me that a non-negligible of residual event is still present in the cloacking region. The authors should zoom on the specific part of the signal for the red curves, typically between 12 and 20.
4. In fig. 3f the authors claim that the signal is "the same" as the even waveform when the cloacking bits are set to 0s. It is in fact hard to say. The authors should fairly compare with a superposition of the different waveforms.
5. The authors succeed in opening a long temporal gap but I don't see the advantage of their device compare to a phase modulator driven by an appropriate electrical signal. Indeed, here the authors have only demonstrated the generation of frequency chirp at the ns time scale which is not a breakthrough performance.
6. Moreover, the authors did not really mentioned the counter-part, which is the giant amount of dispersion they need to open and close their temporal gap. The authors should clarify that point but I guess that they are close to 100 of km of SMF which is not really relevant and compact for applications.
7. Page 14, the eye are not "clear", the authors should moderate their results. Indeed, the noise is very high for 200 Mbit/s eye diagrams.
8. Why there is 2 levels in the eye diagram of fig. 4b?
9. In Fig. 4c, a huge amount of noise is present, please explain why.
10. In Fig. 4c, the level is also very different between ON and OFF so an observer can easily see that something happened to the signal. Please clarify that point.
11. Why the higher level in Fig. 4c is not constant between the 2 measurements, cloak ON/OFF. Explain also the bump in the red curve. This bump should not have been in the "zero slot" of the signal?
12. The chirp is calculated in Fig. 2d, is there any possibility for the authors to support this results by a true measurement?
13. Could they authors extend their technique to more conventional data formats. Dark RZ is not really relevant in nowadays communication process?

Reviewer #3 (Remarks to the Author):

The paper is an experimental implementation of the so-called temporal cloak, first proposed by McCall et al, and first demonstrated by Fridman et al. The key advance presented in the current paper is that the authors have achieved a programmable temporal cloak, in which both the repetition rate and the cloak duration are field programmable.

Insofar as I am able to judge, as a theorist, the robustness and integrity of the experimental methods, the results appear convincing. In Fig. 3 of the paper the authors present clearly the effect of various 'Cloaking bit' patterns on a continuous data-stream, the output having successfully redacted some bits according to the presence/absence of the temporal cloak. The cloak contrast is not high, but the result is still clear and a valid demonstration. The subsequently presented results in Figs. 4 and 5 demonstrate extension of the cloaking window to ns timescales, and the 'stretching' of the cloaking window respectively.

The use of a pre-compensation wave shaper is also shown to enhance the cloaking performance significantly.

The experiment and the results having been successfully described and presented, the question is then whether this represents a significant advance. In my view it is. The first temporal cloaks were limited in both the length of time over which they operated, and were locked into periodic repetition. The current paper overcomes both of these limitations paves the way to wider application of the temporal cloak concept. As such I think the advance is quite significant, and I recommend publication.

Reviewer 1

The author shows a temporal cloaking scheme similar to the split time-lens method with few differences and the main one is that the chirping is done by a tunable micro-resonator fabricated on a silicon wafer. This allows controlling on the opening and the closing of the cloaking in real-time.

The progress of temporal cloaking is fast, and indeed this idea is important and can lead to the implementation of temporal cloaking in telecommunication systems. However, several points need to be addressed for this paper to be ready for publication.

Reply: We thank the reviewer for the positive recommendation for the publication in Nature Communications.

1. There are several problems in the cloaking concept presented here that must be addressed. First, in a temporal cloaking, like the one presented in ref. 14, the system starts with a specific CW wavelength and after the gap opened and closed, it returns to the same wavelength, state of polarization, and mode. Thus, a detector at the end cannot tell that the event happened nor he can tell that there was a cloaking device. Here, the wavelengths do not go back to the original beam. The generated frequency comb is not balanced by an opposite device which changes them beam back to the original wavelength. This is an important point which limits the proposed idea from being implemented in some systems and it must be addressed.

Reply: We thank the reviewer for the professional comments and valuable suggestions. Indeed, the reviewer made a very important and correct point, i.e., the wavelength at the end of cloaking system shall be the same as the original wavelength. Fortunately, this is not a critical issue in practice and this can be straightforward solved by adding a wavelength converter at the end. Because there have been quite a few references of such wavelength converters (Ref. 25 and Ref. 26).

Such a wavelength converter can be inserted after the dispersion compensation to convert multi-wavelength signal to a single wavelength as same as the input light. For example, the output intensity waveform can be converted to a new optical carrier based on cross-gain modulation in semiconductor optical amplifiers. Then the output signal of our system will turn to be continuous wave light and the cloak is both in the time-domain and in the frequency-domain, as a full temporal cloak.

This part is not in the central arena of the current idea, and won't impair or impact any values of our demonstrated devices. Therefore, we add some necessary and succinct discussions on the principle and references regarding the wavelength conversion part, which may be sufficient to let readers have the clear picture of our demonstration plus this accessory of wavelength converter.

Revisions 1#1:

[See **page 7, paragraph 1**]:

Note that the wavelength has changed when the probe light passed through the cloaking system. Thus, this hiding event may be still scouted from the spectral evolution. To solve this issue, a wavelength converter can be inserted after the dispersion compensation to convert multi-wavelength signal to a single wavelength as same as the input light^{25,26}. In this way, the output signal of our system will turn to be a continuous wave light and the cloak could function not only in the time domain but also in the frequency domain, as a full temporal cloak.

[Also see **References and Links, page 20**]:

25. Obermann K, Kindt S, Breuer D, Petermann K. Performance analysis of wavelength converters based on cross-gain modulation in semiconductor-optical amplifiers. *Journal of Lightwave Technology* **16**, 78-85 (1998).
26. Deming L, Hong NJ, Chao L. Wavelength conversion based on cross-gain modulation of ASE spectrum of SOA. *IEEE Photonics Technology Letters* **12**, 1222-1224 (2000).

2. The author must emphasize that when changing the width of the cloaking window, the lengths of the dispersion fibers must be changed as well.

Reply: We thank the reviewer for the valuable comments. The width of cloaking window is in principle relevant to both the length of dispersion fiber (i.e., amount of dispersion) and the sawtooth waveform applied on the ET-MRR (i.e., amount of chirp). In our scheme, we simply change the width of the cloaking window by changing the driving voltage of the ET-MRR. For example, we stretch the cloaking window by changing the repetition rate of sawtooth waveform applied onto ET-MRR (See **Figure 5**). In such a case, the sawtooth waveform should be altered to match the amount of dispersion. The amount of chirp is changed by changing the swept-frequency range, which is determined by the driving voltage of the ET-MRR. For the convenience, we simply change the width of the cloaking window by changing the driving voltage of the ET-MRR while keeping the dispersion fiber at the same length.

Revision 1#2:

[See **page 13, paragraph 1**]:

The reason lies in that the width of the cloaking window is determined by the amount of the chirp of the time lens and the dispersion in our cloaking system. The amount of the chirp of the time lens depends on the driving voltage of the ET-MRR. Thus, we can change the width of the cloaking window by changing the driving voltage of the ET-MRR while keeping the fiber at the same length.

3. I had a hard time understanding the concept. The author does not explain well the working principle of the system. Even after reading the supplementary materials, it took me great effort to fully get it.

a. The author should spend more time explaining the principles of the ET-MRR. In line 59 for example, when the author state "The electrically controllable time lens is enabled by applying an electrical split saw-tooth signal on the ET-MRR and disabled by applying a direct current electrical signal." It should continue with explaining what is the influence of the electric field on the

light. Specifically, I think that it should be: "which acts as a tunable filter were its output wavelength is proportional to the applied voltage"

Reply: We thank the reviewer for the professional comments and valuable suggestions. Following the reviewer's helpful suggestions, we have added to explain the principles of the ET-MRR.

Revision 1#3:

[See **page 3, paragraph 2**]:

The superior time lens consists of an optical frequency comb and an electrically tuned microring resonator (ET-MRR) **acting as a scanning filter, whose output wavelength is proportional to the applied voltage.**

b. A scheme of the ET-MRR is missing with an explanation of what are all the waveguides are for and which waveguide is coupled to which and where. The microscope picture of the device is far from being enough.

Reply: We thank the reviewer for the professional comments and valuable suggestions. Following the reviewer's helpful suggestions, we have added the panorama of the ET-MRR and explanation of the whole structure of the MRR.

Revisions 1#4:

[See **page 5, paragraph 1**]:

The detailed scheme and the whole micrograph of the ET-MRR can be found in Supplementary Note 1.

[Also **Supplementary Material, Supplementary Note 1**]:

Supplementary Note 1: Scheme and whole micrograph of ET-MRR

To implement the cloaking system, electrically tuned microring resonator (ET-MRR) is employed as a tunable filter. The ET-MRR consists of a ring waveguide, two straight waveguides, two contacts and three grating couplers. The global micrograph is shown in Supplementary Fig. 1. In the cloaking

system, the probe beam is vertically coupled into the ET-MRR from the grating coupler on the left and will output from the drop port.

Supplementary Figure 1 | The whole micrograph of ET-MRR.

c. Fig 1(a) and (b) are nice and nowadays it is very common to have a 3d system drawing as is shown in (c) although there is no benefit in such a drawing over a simpler 2d drawing. But the problem is in (d). Since the system is drawn in 3d, all the graphs are tilted which makes them hard to compare and understand. I believe that these graphs should be oriented flat on the page. And maybe even combine several curves in a single graph. For example, curves IV, V, and VI should be on a single graph. While the rest can have a different graph for each curve.

Reply: We thank the reviewer for the professional comments and valuable suggestions. Following the reviewer's helpful suggestions, we have changed the characteristic curves corresponding to the probe light, electrical control signal and event to 2D drawing. And we have combined curves III, IV, V, VI, and VII in a single graph.

Revision 1#5:

[See **page 6, Figure 1**]:

Figure 1 | Schematic and representative of field-programmable silicon temporal cloak. (a) The probe light waveform at the event plane of the traditional temporal cloak is in analog with “hanging curtains”. (b) The “hanging curtains” in our cloaking scheme, featuring a large cloaking window (~ 3 ns) and the cloaking window can be field-programming freely. (c) Schematic of field-programmable silicon temporal cloak. Roman numbers represent various points in the circuit and the characteristic curves corresponding to the probe light, electrical control signal and event are as follows: I, the probe light is continuous wave; II, optical frequency comb generator creates a broadband flat optical frequency comb; III, an ET-MRR driven by an electrical split sawtooth signal and DC signal; IV, the output of drop-port of ET-MRR is split linearly chirped optical signal or monochromatic light, which is required for field-programmable temporal cloak; V, opening gap is achieved while the ET-MRR is driven by a split sawtooth waveform (“cloak on”), and temporal gap does not open while the ET-MRR is driven by a DC signal (“cloak off”); VI, event (dark RZ); VII, the output waveform is CW (“cloak on”) or inverted pulse (“cloak off”). Inset: the microscope images of the fabricated MRR and the zoom-in ring region.

4. Fig. 2(d) should not be drawn with a thin curve since the ET-MRR has a 0.4 nm width, this should also be the width of the curve in the y-direction. This will help in explaining the limitations of the system.

Reply: We thank the reviewer for the professional comments and valuable suggestions. The reviewer is correct. The width of the curve in the y-direction does help in explaining the limitations of the system. For example, the waveform contrast between Cloak ON and Cloak OFF is not very high, which is limited by the bandwidth of the ET-MRR. Following the reviewer's helpful suggestions, we have changed the thin curve (Fig. 2(d)) to a curve with a certain width in the y-direction.

Revisions 1#6:

[See page 9, Figure 2]:

Figure 2 | Electrically controllable silicon-based photonic time lens. (a) Measured transmission spectra at the drop-port of the fabricated ET-MRR with different DC voltages applied on the microelectrode. (b) The output spectrum of the optical frequency comb generator. Inset: original flat optical frequency comb (green) and optical filter of wave shaper (purple). (c) Driving electrical signal (mixed split sawtooth signal and DC signal) of the ET-MRR. (d) The time-resonance wavelength curve of the time lens is calculated.

[Also see page 9, paragraph 1]:

Note that the finite width of the curve in the y-direction is attributed to the bandwidth of the ET-MRR.

5. Can the author suggest a way to fabricate the entire cloaking system on a silicon chip?

Reply: We thank the reviewer for the interesting and valuable suggestion. In the paper, we have explained the potential of our temporal cloaking system to be fully chip-integration from four aspects. To fabricate the entire temporal

cloaking system on a chip, an integrated high-speed optical intensity modulator is required, which acts as the event emulator. Currently, the integrated high-speed optical intensity modulator is mature in SOI fabrication foundry.

Revisions 1#7:

[See **page 17, paragraph 1**]:

Finally, to fabricate the entire cloaking system on a silicon chip, an on-chip high-speed intensity modulator is required, which acts as the event emulator. Currently, the integrated high-speed intensity modulator is mature in SOI fabrication foundry⁴²⁻⁴⁴, which motivates a promising future for the full integration.

[Also see **References and Links, page 22**]:

44. Xiao X, *et al.* High-speed, low-loss silicon Mach–Zehnder modulators with doping optimization. *Opt Express* **21**, 4116-4125 (2013).
45. Rao A, *et al.* High-performance and linear thin-film lithium niobate Mach–Zehnder modulators on silicon up to 50 GHz. *Opt Lett* **41**, 5700-5703 (2016).
46. He M, *et al.* High-performance hybrid silicon and lithium niobate Mach–Zehnder modulators for 100 Gbits–1 and beyond. *Nature Photon*, (2019).

Once these comments are addressed, I believe that this manuscript can be accepted for publication.

Reply: We thank the reviewer for the positive recommendation for the publication in Nature Communications.

Thanks again for the valuable comments and helpful suggestions.

Reviewer 2

The paper from Zhou and coworkers deals with the experimental demonstration of temporal cloaking based on the technique of time lens. In their work, they used a novel device made of a programmable electrically tuned micro-ring resonator to create the chirp profile involved in the time lens operation. The paper is well written and provide a pedagogical approach. However, to my opinion, the novelty and quality of the results are not good enough to deserve publication in a high impact journal, in particular Nature Communications. Consequently, I reject this paper and encourage We to submit their work elsewhere in a less impact journal after the following Revision.

1. The concept and physical effects are not new and was already demonstrated in various publications, in particular in [14,15,21] but not only.

Reply: We thank the reviewer for the valuable comments. The concept and physical effects of the “temporal cloak” was already demonstrated in various publications (Gaeta et al., Nature 481, 62-65, 2012; Weiner et al., Nature 498, 205-208, 2013; Wong KK et al. Optics Letters 42, 767-770, 2017). However, until now the state-of-the-art cloaking experiments exhibited only a fixed and small cloaking window with picosecond-level due to the periodicity and aperture limit of time lens, as shown in Fig. 1a. Thus, the periodical and small cloaking window (<200ps) hinders its applications in data shield and secure communication. And, it is more powerful and practical to make the cloaking window field-programmable (i.e., the cloaking window can be switched off, switched on, and stretched freely, as shown in Fig. 1b) since different types of optical packets can be hidden freely with the cloaking system.

To address this issue, our manuscript reports on a major breakthrough to make the temporal cloak programmable, shown in Fig. 1b. We demonstrate a field-programmable temporal cloak with a record cloaking window of nanosecond-level benefiting from a unique electrically controllable silicon-based time lens (experimental setup is shown in Fig.1c). We believe that we make a breakthrough and have high impacts in the following aspects:

1) We demonstrate, for the first time, a field-programmable temporal cloak with potential applications in data shield, enabling to share some public data to the user but conceal other private data in real time. Importantly, we break the periodicity of cloaking window. Previous demonstration of cloaking system had no such powerful functions.

2) With dynamic control of the driving electrical signals on the microring, our cloaking window could be stretched and switched in real time from 0.449 ns to 3.365 ns, 17 times larger than the best results reported so far.

3) Our unique silicon-based time lens has distinct advantages of field-programmable cloaking window, moderate power consumption and potential for photonic integration. We suggest that the field-programmability of temporal cloak may make its applications, such as secure communication and data compression, more practical and closer to our daily life.

From the above, our field-programmable temporal cloak is novelty, unprecedented performance, and the broad implication. Thus, we think it has high impact, and deserves to be published in Nature Communications.

2. The only novelty is the implementation of a programmable silicon micro-ring. However, the time response appears very low compare to phase modulator. It is probably the reason why We focus the paper on the record temporal gap they succeed to open. The advantage of their device is the programmable nature but it is not very clearly demonstrated to my opinion. To really support their claims, We should have demonstrate some cloaking with arbitrary sequences and duration of the cloaking window, not only some periodic train as provided by figs. 4 and 5. Indeed, the demonstration and advantage of this programmable device is almost absent of this paper, except in figs. 3c-e but for periodic dark event. Moreover, We should have demonstrated the cloaking of data packets at high repetition rate rather than very low repetition rate data, 200Mbit/s is very weak. Indeed, the advantage of generating long temporal gap could be to hide long data packet, not only a ns pulse.

Reply: We thank the reviewer for the professional comments and valuable suggestions. We feel fortunate and honored to see the reviewer's comments also appraised of our temporal cloak being novel because of the field-programmable feature. In principle, the time response of the field-programmable temporal cloak depends on the time response of the swept-frequency filter. The swept-frequency filter of ET-MRR can be designed for either high-speed or low-speed applications. Therefore, we can not only create the temporal cloak with the high-speed and small width of cloaking window, but also achieve the temporal cloak with low-speed and large width of cloaking window. The high-speed cloaking rate and the large width of cloaking window are natively contradicting each other, and it is difficult and almost forbidden to obtain at the same time. Some applications need to achieve high-speed temporal cloak (Weiner et al., Nature 498, 205-208, 2013; J. Fatome et al. Nature communications 5, 4678, 2014; Wong KK et al. Optics Letters 42, 767-770, 2017), but it is equally challenging to obtain large width of cloaking window. And the field-programmable temporal cloak is even more challenging. In our scheme, we focus on the cloaking system on the record temporal gap. And the unique advantage of our system is the filed-programmable nature with decently enhanced width up to ns window, as a fair progress made compared to the previous/milestone ps and static temporal cloaks.

The reviewer's suggestion to show arbitrary sequences and duration of the cloaking window is very constructive for our manuscript. To really support our claims of field programmability, we have performed more experimental data, such as periodic event controlled by periodic cloaking bits [**See Figure 3(a, d)**], periodic event controlled by arbitrary cloaking bits [**See Figure 3(b, c)**], arbitrary event controlled by arbitrary cloaking bits [**See Figure 3(e)**], and arbitrary event controlled by periodic cloaking bits [**See Figure 3(f)**]. These effective demonstrations can prove that our cloaking system has a breakthrough performance in dynamic field-programmability. We also demonstrate that both two-pulse event and one-pulse event can be hidden by the cloaking bits [**See Figure 3(f)**]. This shows the promising advantage of using long temporal gap to hide more pulses, not just a ns pulse. In the revised

paper, following the reviewer's helpful suggestions, we carefully add those new experimental data to remove the concerns of this reviewer.

Revisions 2#1:

[See page 11, Figure 3]:

Figure 3 | Field-programmable temporal cloak. (a) The DC waveform (red) with a low ripple factor is received when the cloak switch (cyan) and events (blue) are turned on all the time. (b,c) The received output waveforms (red) with high-level signal at the cloaking bits when the events (blue) are all turned on and the cloak switch (cyan) is selectively turned on. (d) The high-contrast voltage (red) output waveform when the cloak switch (cyan) is turned off and the events (blue) are turned on all the time. (e) **Arbitrary sequence temporal cloak for hiding periodic sequence dark events with half cloaking repetition rate.** (f) **Periodic temporal cloak for hiding arbitrary sequence dark events.** The gray labels represent the cloaking bits.

[Also see page 10, paragraph 1]:

Figure 3 shows some typical experiment results of field-programmable temporal cloak by varying the cloaking bits on the ET-MRR. The first row

shows the sequence of cloaking bits at the bit rate of 200 Mbit/s. The second row shows a user-defined sequence event (blue) with dark RZ signals.

[Also see **page 10, paragraph 1**]:

Further, we set the switching bits to random sequences such as “1010011000”, and select 100 MHz dark RZ signal as the event. As shown in Fig. 3e, the periodic sequence dark events with half cloaking repetition rate is selectively cloaked by the arbitrary sequence temporal cloak. In addition, an arbitrary sequence dark events with both two-pulse event and one-pulse event are erased by the periodic temporal cloak, as shown in Fig. 3f. It shows that the long cloaking window could hide more pulses, not only a single pulse.

3. The way We qualified the cloaking efficiency is quite questionable. An extinction ratio between ON and OFF should be better. 2 digits is also unnecessary. Moreover, We used a very large scale to present their results of cloaking. In fig. 3 for instance, it is difficult to see and fairly judge the efficiency of the process. It seems to me that a non-negligible of residual event is still present in the cloaking region. We should zoom on the specific part of the signal for the red curves, typically between 12 and 20.

Reply: We thank the reviewer for the valuable comments. The extinction ratio between ON and OFF is not high. The main reason is that the low signal-to-noise ratio of the link results in a low extinction ratio. In our scheme, the electrically controllable silicon-based time lens with a swept-frequency filter introduces a high loss for the output light signal, and the output light with some noise is filtered by the swept-frequency filter with 0.15 nm bandwidth. Thus, the signal-to-noise ratio is not high. However, comparing Figs. 3(a) and 3(d), the distinct waveform difference can be clearly observed between states of “Cloak ON” and “Cloak OFF”. In the meanwhile, as Reviewer 3 said *“In Fig. 3 of the paper, we present clearly the effect of various ‘Cloaking bit’ patterns on a continuous data-stream, the output having successfully redacted some bits according to the presence/absence of the temporal cloak. The cloak contrast is not high, but the result is still clear and a valid demonstration.”* Thus, the way

we qualified the cloaking efficiency should be reasonably valid. The extinction ratio can be improved further by optimizing the DCF length, link loss, and the electrical signal on ET-MRR. Nevertheless, that could be a follow-up improvement but not a central breakthrough point.

[See **page 11, paragraph 1**]:

“We may notice that the waveform contrast between cloak on and cloak off is not very high, which can be improved further by optimizing the DCF length, link loss, and the electrical signal on ET-MRR.”

4. In fig. 3f We claim that the signal is “the same” as the even waveform when the cloaking bits are set to 0s. It is in fact hard to say. We should fairly compare with a superposition of the different waveforms.

Reply: We thank the reviewer for the valuable comments. The reviewer is correct. When the cloaking bits are set to 0s, the output signal is broadened after the dispersion compensating fiber. The same issue also exists in the reference (Weiner et al., Nature 498, 205-208, 2013). As a remedy, we can add a spool of single mode fiber to compensate the dispersion. When the cloaking bits are set to 1s, the added fiber does not affect the output light signal with a single wavelength.

Revision 2#2:

[See **page 10, paragraph 1**]:

The output signal is broadened after the DCF and it can be compensated by introducing a spool of SMF to compensate the dispersion of the DCF.

5. We succeed in opening a long temporal gap but I don't see the advantage of their device compare to a phase modulator driven by an appropriate electrical signal. Indeed, here We have only demonstrated the generation of frequency chirp at the ns time scale which is not a breakthrough performance.

Reply: We thank the reviewer for the comments. It is required to generate the long cloaking window for hiding long-duration event. The high-speed cloaking rate and the large width of cloaking window are opposite in trend, so it is extremely difficult to obtain at the same time. Thus, the temporal cloak with the large cloaking window can only operate at low rate, and hide a long packet. On the other hand, the temporal cloak based on the phase modulator can be operate at telecommunication data rate and hide the high-speed data stream. However, its cloaking window is limited by the time lens with small time aperture based on the phase modulator. Those two groups fit different application scenarios, and hence it is not quite fair to compare in the way of this reviewer. Our work focuses on the long cloaking window temporal cloak.

Moreover, the temporal cloak based on the phase modulator exhibited only a fixed cloaking window due to the periodicity of the time lens. In our manuscript, we demonstrate a field-programmable temporal cloak benefiting from an electrically controllable silicon-based time lens. It is more powerful and practical to make the cloaking window field-programmable (namely, the cloaking window can be switched off, switched on, or stretched freely). Then, different types of optical packets can be hidden freely with one cloaking system. The field-programmable temporal cloak can be widely used in data shield and secure communication. Thus, our manuscript reports on a breakthrough to make the temporal cloak programmable, and the ns scale cloaking window is also a breakthrough performance, fairly benchmarked against those milestone works in its own kind.

In particular, our system can hide both ns-level pulse event (See Fig. 4c) and two-pulse event with ps-level (See Fig. 3f). And the cloak sequences can be set arbitrarily. Note that the cloaking system based on phase modulator is impossible to do so.

6. Moreover, We did not really mentioned the counter-part, which is the giant amount of dispersion they need to open and close their temporal gap. We should clarify that point but I guess that they are close to 100 of km of SMF which is not really relevant and compact for applications.

Reply: We thank the reviewer for the professional comments and valuable suggestions. As shown in Supplementary Fig. 5, an 80 km single mode fiber with a total dispersion of 1340.5 ps/nm is used to open the cloaking window. Another 10.3 km dispersion compensating fiber with a total dispersion of -1342 ps/nm is used to close the cloaking window. In the revised paper, following the reviewer's helpful suggestions, we carefully take these concerns raised by the reviewer into full consideration, and added the sentences in the revised supplementary information.

The reviewer mentioned that the long fiber is not relevant and compact for applications. But our scheme with an ET-MRR is the key step towards the full integration of cloaking system on a chip. We thereby showcase a perspective for the photonic integration of cloaking system with an electrically controllable swept-frequency time lens from four aspects. We firmly believe that the breakthrough of integrated devices with large dispersion will greatly promote the development of the entire cloaking system on a chip.

Revision 2#3:

[See **Supplementary Material, Supplementary Note 5**]:

An 80 km single mode fiber with a total dispersion of 1340.5 ps/nm is used to open the cloaking window. Another 10.3 km dispersion compensating fiber with a total dispersion of -1342 ps/nm is used to close the cloaking window.

7. Page 14, the eye are not "clear", We should moderate their results. Indeed, the noise is very high for 200 Mbit/s eye diagrams.

Reply: We thank the reviewer for the valuable comments. According to the experimental results, it is known that the 200 Mbit/s eye pattern is clearer than the 400 Mbit/s and 800 Mbit/s eye patterns. The reason is that the time response of the ET-MRR is limited. It will cause the aberration of the time lens with an operation speed higher than 200 Mbit/s. The larger the aberration is, the less clear the eye is. Thus, the 400 Mbit/s and 800 Mbit/s eye are not clear than the 200 Mbit/s eye. Besides, the larger the aberration, the worse the cloaking window. As shown in Fig. 4a and Figs. 5a-b, the 200 Mbit/s cloaking

window is better than the 400 Mbit/s and 800 Mbit/s cloaking window. Nevertheless, that could be a follow-up improvement with a large-bandwidth ET-MRR but not impair or impact any values of our demonstrated devices. In the revised paper, following the reviewer's helpful suggestions, we seriously and carefully take the reviewer's professional comments into full consideration.

Revision 2#4:

[See **page 14, paragraph 1**]:

Note that the eye patterns for bit rate of 400 MHz and 800 MHz are worse than that of 200 MHz, because the aberration of time lens is more serious at higher operation speed, limited by the time response of the ET-MRR.

8. Why there is 2 levels in the eye diagram of fig. 4b?

Reply: We thank the reviewer for the valuable comments and suggestions. As shown in Fig. 4b, there are 2 levels in the eye diagram, caused by the electrical arbitrary waveform generator (Keysight M8195A, EAWG), which is not calibrated in the experiments.

Revision 2#5:

[See **page 12, paragraph 1**]:

There are 2 levels in the eye diagram, caused by the electrical arbitrary waveform generator (Keysight M8195A) without calibration.

9. In Fig. 4c, a huge amount of noise is present, please explain why.

Reply: We thank the reviewer for the professional comments. The reviewer is correct. As shown in Fig. 4c, a huge amount of noise is present, and the measured eye diagram shows a thick eyelid which is caused by the high loss of the fiber link and photonic chip, as well as the fiber amplifier noise.

[See page 12, paragraph 1]:

“The measured eye diagram shows a thick eyelid which is caused by the high loss of the fiber link and photonic chip, as well as the fiber amplifier noise.”

10. In Fig. 4c, the level is also very different between ON and OFF so an observer can easily see that something happened to the signal. Please clarify that point.

Reply: We thank the reviewer for the professional comments. The reviewer is correct. It is extremely challenging to achieve the consistent level of the cloak on and off. “The voltage levels in the cloaked cases do not align with the uncloaked peaks because of amplifier saturation, which forces the average output power to remain constant; a continuous-wave waveform must drop slightly compared to the modulated case to conserve the integrated power” (referred to Weiner et al. Nature 498, 205-208, 2013, see page 208, paragraph 2, line 9). In the revised paper, following the reviewer’s helpful suggestions, we seriously and carefully take the reviewer’s professional comments into full consideration.

Revision 2#6:

[See page 12, paragraph 1]:

Note that the level is also different between cloak on and off because of amplifier saturation of the EDFA, which keeps the average output power constant. Thus, the CW waveform drops slightly compared to the modulated case.

11. Why the higher level in Fig. 4c is not constant between the 2 measurements, cloak ON/OFF. Explain also the bump in the red curve. This bump should not have been in the “zero slot” of the signal?

Reply: We thank the reviewer for the valuable comments. There are two main reasons for the nonconstant higher level and the bump. Firstly, the chirp of the time lens is practically difficult to 100% match with normal dispersion and

dispersion compensation. Secondly, the wave shaper is hard to match to completely remove the bump. In the revised paper, following the reviewer's helpful suggestions, we carefully take the reviewer's professional comments into consideration.

Revision 2#7:

[See **page 12, paragraph 1**]:

At the same time, the non constant higher level and the ripple is caused by the mismatch between the chirp of the time lens and the dispersion devices.

12. The chirp is calculated in Fig. 2d, is there any possibility for We to support this results by a true measurement?

Reply: We thank the reviewer for the valuable comments. The reviewer is correct. The chirp is a very important parameter for the time lens. But the chirp is hard to measure directly. Thus, we measure the chirp indirectly by calculating the instantaneous frequency of the output signal of the ET-MRR according to the characteristic curve of wavelength shift of the ET-MRR and the driven electrical signal on the ET-MRR. And the characteristic curve of wavelength shift and the driven electrical signal are measured. Thus, the indirect measurement of the chirp is valid and reasonable.

13. Could they authors extend their technique to more conventional data formats. Dark RZ is not really relevant in nowadays communication process?

Reply: We thank the reviewer for the professional comments and valuable suggestions. As Weiner et al explained in their work (Weiner et al. Nature 498, 205-208, 2013), they used a dark RZ modulation format, which ensures that the optical transmission function returns to a maximum during each cycle, as required to provide temporal regions through which the compressed probe pulses can pass. Thus, most temporal cloak schemes are demonstrated by hiding the dark RZ data. In fact, the cloaked event can also be other types of

on-off keying (OOK) signals, which can be first converted into dark RZ signals by some transformations of format converters ^[1].

[1] J. Dong, X. Zhang, S. Fu, J. Xu, P. Shum, and D. Huang, "Ultrafast all-optical signal processing based on single semiconductor optical amplifier and optical filtering," *Journal of Selected Topics in Quantum Electronics* **14**, 770-778 (2008).

Thanks again for the valuable comments and helpful suggestions.

Reviewer 3

The paper is an experimental implementation of the so-called temporal cloak, first proposed by McCall et al, and first demonstrated by Fridman et al. The key advance presented in the current paper is that We have achieved a programmable temporal cloak, in which both the rep-rate and the cloak duration are field programmable.

In so far as I am able to judge, as a theorist, the robustness and integrity of the experimental methods, the results appear convincing. In Fig. 3 of the paper We present clearly the effect of various 'Cloaking bit' patterns on a continuous data-stream, the output having successfully redacted some bits according to the presence/absence of the temporal cloak. The cloak contrast is not high, but the result is still clear and a valid demonstration. The subsequently presented results in Figs. 4 and 5 demonstrate extension of the cloaking window to ns timescales, and the 'stretching' of the cloaking window respectively.

The use of a pre-compensation wave shaper is also shown to enhance the cloaking performance significantly.

The experiment and the results having been successfully described and presented, the question is then whether this represents a significant advance. In my view it is. The first temporal cloaks were limited in both the length of time over which they operated, and were locked into periodic repetition. The current paper overcomes both of these limitations paves the way to wider application of the temporal cloak concept. As such I think the advance is quite significant, and I recommend publication.

Reply: We thank the reviewer for the positive recommendation for the publication in Nature Communications.

Thanks again for the valuable comments and helpful suggestions.

REVIEWERS' COMMENTS:

Reviewer #1 (Remarks to the Author):

The author has answered all my comments and the paper was much improved. I also read the comments and responses for the other reviewers and although I do not agree with all their comments the author answered all their comments as well.

I do think that this paper is novel enough and will interest the reader thanks to its unique use of microcavity for this tunable cloaking scheme.

Therefore, I believe that this manuscript is ready to be published.

Moti Fridman

Reviewer #2 (Remarks to the Author):

This new version of the paper from Zhou and coworkers did not convince me at all. The changes are only substantial. The authors agree with numerous of my comments but most of the points are only poorly discussed in the answer and a few comments have been included in the revised version.

See for instance, but not only, answers to points 3, 4, 7, 8, 9, 10 etc.

Point 3, the authors did not improve the clarity of the figure. It is still difficult to judge the quality of their results. Same for point 4. Point 6, the authors believe future development... But using an integrated device surrounded by 100km of fibers is for me inappropriate. 7, 8, 9, 10, 11 the authors just agree that the performance are quite poor and that's it. Response to point 1 of the first reviewer is also only justified by 2 new references. However, this point is fundamental. The change of the central wavelength is a true issue for this process and incorporating a wavelength converter is not just straightforward.

The authors have also published a paper on MRR. They should explain the novelty of their new device/design.

Aoling Zheng, Opt. Lett. 39, 6355-6358 (2014)

The only true improvement of this revised version is figure 3f. However, it is hard to see the quality of these results because of the scale chosen by the authors.

This paper does not meet the standard of publication for nature communication. It is too technical, and the claim are to my opinion poorly supported by the experimental results. In conclusion, the quality of the results are not good enough to deserve publication in a high impact journal, in particular Nature Communications. Consequently, I reject this paper and encourage the authors to submit their work elsewhere in a less impact journal, and more technical, like JLT.

Reviewer #3 (Remarks to the Author):

I note the many improvements made to the manuscript that address the concerns of the other reviewers.

As I said in my first review, I think the success of this manuscript depends on whether the demonstration of a field-programmable (temporal) cloaking window is sufficiently important to merit its publication in a high-profile journal. I re-iterate that in my view it is, and I think the authors have done a good job of reinforcing their case in their Reply [cf. points 1)-3) on p.11].

I recommend publication.

REVIEWERS' COMMENTS:

Reviewer #1 (Remarks to the Author):

The author has answered all my comments and the paper was much improved. I also read the comments and responses for the other reviewers and although I do not agree with all their comments the author answered all their comments as well.

I do think that this paper is novel enough and will interest the reader thanks to its unique use of microcavity for this tunable cloaking scheme.

Therefore, I believe that this manuscript is ready to be published.

Reply: We thank the reviewer for the positive recommendation for the publication in Nature Communications.

Thanks again for the valuable comments and helpful suggestions.

Reviewer #2 (Remarks to the Author):

This new version of the paper from Zhou and coworkers did not convince me at all. The changes are only substantial. The authors are agree with numerous of my comments but most of the points are only poorly discussed in the answer and a few comments have been included in the revised version.

See for instance, but not only, answers to points 3, 4, 7, 8, 9, 10 etc.

Point 3, the authors did not improve the clarity of the figure. It is still difficult to judge the quality of their results. Same for point 4. Point 6, the authors believe future development...

Reply: We thank the reviewer for the comments. As we mentioned in the manuscript and former response letter, the extinction ratio between ON and OFF is not high. The main reason is that the low signal-to-noise ratio of the link results in a low extinction ratio. In our scheme, the electrically controllable silicon-based time lens with a swept-frequency filter introduces a high loss for the output light signal. When the broadband optical frequency comb propagates through the swept-frequency filter and the selected light signal with very low power is amplified by the erbium doped fiber amplifier (EDFA), and some noise is introduced. Thus, the signal-to-noise ratio is not high. Nevertheless, it does not impair the key claims and demonstration of advanced functionalities (i.e., programmable cloaking window) compared to the previous temporal cloaks.

[See **page 11, paragraph 1**]:

We may notice that the waveform contrast between *cloak on* and *cloak off* is not very high. In our scheme, the electrically controllable silicon-based time lens with a swept-frequency filter introduces a high loss for the output light

signal. When the broadband optical frequency comb propagates through the swept-frequency filter and the selected light signal with very low power is amplified by the erbium doped fiber amplifier (EDFA), some noise is introduced. Thus, the signal-to-noise ratio is not high. The waveform contrast can be improved further by optimizing the DCF length, link loss, and the electrical signal on ET-MRR.

But using an integrated device surrounded by 100km of fibers is for me inappropriate.

Reply: We thank the reviewer for the comments. In our field-programmable temporal cloaking system, only the scanning filter is an integrated device and the others are discrete devices. It is a widely adopted proof-of-concept approach to introduce a large dispersion, using a long single-mode fiber (See Refs. 14, 15, 21). Indeed, it is still a big challenge to realize large dispersion with an integrated chip. But probably in the near future, the integrated chip with large dispersion could be implemented thus fully integration is possible. The discussion of fully integration is also addressed in Discussion (See **page 17, paragraph 2**).

7, 8, 9, 10, 11 the authors just agree that the performance are quite poor and that's it.

Reply: We thank the reviewer for the comments. We request the respected referee to benchmark using a fair reference. Whether it is super-excellent or

poor shall be compared to the state-of-the-art in its kind (apple-to-apple), instead of apple-to-pear comparison.

First, the details of the system performances, such as eye patterns, noise performance, and waveform contrast ratio, have been addressed according to the reviewer's comments in the revised manuscript.

Second, as Reviewer #3 said "... The cloak contrast is not high, but the result is still clear and a valid demonstration." Therefore, our main breakthrough is the first demonstration of a programmable-length cloaking window rather than performance improvements. A follow-up improvement scheme could be further made, but that is rather not a central breakthrough point.

Third, even if we compare the contrast ratio between *cloak on* and *cloak off*, we can find that in our case of periodic cloaking window, the ratio is about 21.00%, comparable to Refs. 14, 15, 21, which is about 20.90%, 27.59%, 38.89%, respectively. Note that the contrast ratio between *cloak on* and *cloak off* is defined as the peak-to-peak voltage of output signals in the state of *cloak on* over the peak-to-peak voltage of output signals in the state of *cloak off*. And the smaller the contrast ratio, the better the cloaking performance. From the above comparison, the contrast ratio of our cloaking system is less than most of previously reported cloaking systems. Thus, our cloaking performance is superior.

Response to point 1 of the first reviewer is also only justified by 2 new references. However, this point is fundamental. The change of the central wavelength is a true issue for this process and incorporating a wavelength converter is not just straightforward.

Reply: We thank the reviewer for the comments. Our research group has focused on all-optical wavelength conversion for many years. This is not a critical issue in practice and this can be straightforwardly solved by adding a wavelength converter at the end. Such wavelength converters (Refs. 25 and Ref. 26) are quite standard and mature, and this issue is true but it has been easily solved before. Hence, we try not to repeat every piece of classic wisdom in one work.

The authors have also published a paper on MRR. They should explain the novelty of their new device/design.

Aoling Zheng, *Opt. Lett.* 39, 6355-6358 (2014)

Reply: We thank the reviewer for the comments. Our tunable cloaking scheme benefits from its unique use of the ET-MRR. In the previous *Opt. Lett.* paper, we demonstrated photonic differentiator with an MRR. In this manuscript, we focus on a new type of temporal cloak, that is the field-programmable temporal cloak. They are totally different applications.

The only true improvement of this revised version is figure 3f. However, it is hard to see the quality of these results because of the scale chosen by the authors.

Reply: We thank the reviewer for the valuable comments. Figure 3f shows that both two-pulse event and one-pulse event can be hidden by the cloaking bits. This showcases the promising advantage of using long temporal gap to hide more pulses, not just a ns pulse. The result, though not perfect, is still clear with valid demonstrations. We present clearly the effect of various 'Cloaking

bit' patterns on a continuous data-stream, the output has successfully redacted some bits according to the presence/absence of the temporal cloak.

This paper does not meet the standard of publication for nature communication. It is too technical, and the claim are to my opinion poorly supported by the experimental results. In conclusion, the quality of the results are not good enough to deserve publication in a high impact journal, in particular Nature Communications. Consequently, I reject this paper and encourage the authors to submit their work elsewhere in a less impact journal, and more technical, like JLT.

Reply: We thank the reviewer for the comments. But we disagree with your comments.

We solve the long-held critical challenges for temporal cloaks. Temporal cloaks, which hide a temporal event within a signal, have been previously limited to very short and periodic event cloaking. Thus, we report a temporal cloak with a programmable-length cloaking window using a silicon microring and optical frequency comb. We break the periodicity of cloaking window and propose a new concept of the field-programmable temporal cloak. The programmable-length of temporal cloak is experimentally demonstrated. Previous demonstrations of cloaking system had no such powerful functions. Although the extinction ratio between *cloak on* and *cloak off* is not ultra-high, the experimental results can unanimously prove that our temporal cloak has the characteristic of the field-programmability.

Thanks again for the valuable comments and helpful suggestions.

Reviewer #3 (Remarks to the Author):

I note the many improvements made to the manuscript that address the concerns of the other reviewers.

As I said in my first review, I think the success of this manuscript depends on whether the demonstration of a field-programmable (temporal) cloaking window is sufficiently important to merit its publication in a high-profile journal. I re-iterate that in my view it is, and I think the authors have done a good job of reinforcing their case in their Reply [cf. points 1)-3) on p.11].

I recommend publication.

Reply: We thank the reviewer for the positive recommendation for the publication in Nature Communications.

Thanks again for the valuable comments and helpful suggestions.